# How Biology Guides the Combination of Locoregional Interventional Therapies and Immunotherapy for Hepatocellular Carcinoma: Cytokines and Their Roles

**DOI:** 10.3390/cancers15041324

**Published:** 2023-02-19

**Authors:** Yan Fu, Chu Hui Zeng, Chao An, Yue Liu, Ji Hoon Shin, Xiao Li

**Affiliations:** 1Department of Interventional Therapy, National Cancer Center/National Clinical Research Center for Cancer/Cancer Hospital, Chinese Academy of Medical Sciences and Peking Union Medical College, Beijing 100021, China; 2Department of Radiology and Research Institute of Radiology, Asan Medical Center, University of Ulsan College of Medicine, Seoul 05505, Republic of Korea; 3Department of Interventional Ultrasound, Chinese PLA General Hospital, 28 Fuxing Road, Beijing 100853, China

**Keywords:** locoregional interventional therapies, hepatocellular carcinoma, immune checkpoints, immunotherapy, cytokines

## Abstract

**Simple Summary:**

Unresectable hepatocellular carcinoma (HCC) is the main type of primary liver cancer and poses a challenge to the healthcare system across the world. Immune checkpoint inhibitor (ICI)-based immunotherapy has become a recent focus of HCC treatment. However, its high risk of treatment-related severe adverse events makes effective combination strategies to lower toxicity and improve clinical outcomes an urgent need. Although locoregional interventional therapies are considered promising strategies to synergize ICI-based immunotherapies by promoting the release of tumor antigens and proinflammatory cytokines, current clinical trials show controversial results. Since cytokines play critical roles in the combination therapy of LITs and immunotherapy, this review aims to summarize the biological roles of cytokines and their therapeutic potentials in the LITs combined with ICI-based immunotherapies.

**Abstract:**

As most patients with hepatocellular carcinoma (HCC) are diagnosed at the intermediate or advanced stage and are no longer eligible for curative treatment, the overall survival rate of HCC remains unsatisfactory. Locoregional interventional therapies (LITs), and immune checkpoint inhibitor (ICI)-based immunotherapy, focus on treating HCC, but the efficacy of their individual application is limited. Therefore, the purpose of this review was to discuss the biological roles of cytokines and their therapeutic potential in the combination therapy of LITs and ICI-based immunotherapy. The two common techniques of LITs are ablative and transarterial therapies. Whether LITs are complete or incomplete can largely affect the antitumor immune response and tumor progression. Cytokines that induce both local and systemic responses to LITs, including interferons, interleukins, chemokines, TNF-α, TGF-β, VEGF, and HGF, and their roles are discussed in detail. In addition, specific cytokines that can be used as therapeutic targets to reduce immune-related adverse events (irAEs) are introduced. Overall, incomplete LITs in a tumor, combined with specific cytokines, are thought to be effective at improving the therapeutic efficacy and reducing treatment-induced irAEs, and represent a new hope for managing unresectable HCC.

## 1. Introduction

Primary liver cancer, which is ranked as the sixth most diagnosed cancer and the third most common cause of cancer-related death, remains one of the challenging healthcare issues worldwide [1]. Hepatocellular carcinoma (HCC) is the main type of primary liver cancer, accounting for approximately 90% of all cases [2,3]. Several risk factors for HCC include chronic hepatitis B virus (HBV) infection, chronic hepatitis C virus (HCV) infection, chronic alcohol consumption, metabolic syndrome, and smoking [2,4,5]. Despite the significantly increased incidence of HCV infection in western countries, chronic HBV infection is still the most common risk factor for HCC, accounting for about 54% of cases worldwide [2,4,5]. The main reason for the unsatisfactory overall survival (OS) rate (5-year: 18%) of HCC is that most patients are diagnosed at the intermediate or advanced stage, losing the opportunity for curative treatment [2,5,6]. The prognosis varies widely at different disease stages. For patients at the very early and early stages, the median OS is expected to be ≥5 years after curative treatment [5,7]. However, the expected median OS is much lower at the intermediate, advanced, and terminal stages, which are 2.5 years, 2 years, and 3 months, respectively [5,7]. Therefore, more appropriate treatment protocols to improve the clinical outcomes of unresectable HCC are needed.

In the last decade, the treatment regimen of advanced-stage HCC has been largely updated by the appearance of the immune checkpoint inhibitor (ICI)-based immunotherapy [7]. Immune checkpoint proteins (ICPs), also known as inhibitory immune molecules and gate-keepers of the immune response, are upregulated by the activated immune cells to prevent inappropriate immune response and maintain peripheral tolerance [8,9,10]. However, tumor cells also use these ICP-dependent immunoinhibitory pathways to suppress the antitumor immune response and promote their evasion from immune surveillance [8,11]. Therefore, targeting ICPs to renormalize the antitumor response is considered a promising therapeutic strategy for cancer [8]. The ICPs mainly include PD-1, PD-L1, CTLA-4, LAG-3, TIM-3, TIGHT, and BTLA [11]. Among them, anti-PD-L1 antibodies and anti-CTLA-4 antibodies are the emphases of the current clinical application for HCC [10]. However, the monotherapies of anti-PD-1/PD-L1 or anti-CTLA-4 antibodies failed to prolong OS of advanced-stage HCC in several phase III randomized controlled trials, with objective response rates (ORRs) ranging from 15% to 20% [12,13,14]. Although the combined immunotherapies, including dual ICI treatment or ICIs combined with anti-angiogenic tyrosine kinase inhibitors (TKIs), can increase ORR to 30–36% and prolong OS to 19 months in the advanced-stage HCC, the risk of severe treatment-related adverse events (≥grade 3) is relatively higher, up to 67% [12,15,16,17,18]. Effective combination strategies with low toxicity are required to improve the clinical outcomes of ICI-based immunotherapies for HCC.

Locoregional interventional therapies (LITs), defined as imaging-guided minimally invasive procedures to directly treat diseases, are believed to be a promising option to synergize ICI-based immunotherapies [19]. With lower risk and faster recovery from procedures, LITs are considered an essential part of HCC treatment, and approximately 50–60% of patients with HCC are treated with LITs [2,5,19]. They use local ablative and transarterial techniques to eliminate or reduce the viability of tumor cells, delaying tumor progression and increasing OS [19,20]. Since LITs can stimulate the antitumor immune response by promoting the release of tumor antigens and proinflammatory cytokines [19,20], several preclinical and clinical studies have been launched to investigate the safety and efficacy of the combination treatment of LITs and ICIs. However, the efficacy is controversial based on the current real-world retrospective studies. In particular, the two-edged-sword effects of LITs on modulating the local and systemic immune response may further influence the efficacy of ICI-based immunotherapies. Since a better understanding of the underlying biology is critical for treatment decision-making and improving clinical outcomes, this review aims to discuss the biological roles of cytokines and their therapeutic potential in the combination therapies of LITs and ICI-based immunotherapies.

## 2. Locoregional Interventional Therapies in Combination with ICI-Based Immunotherapy for HCC: Opportunities and Challenges

### 2.1. Ablative Therapies

Ablative therapies are considered a potentially curative approach for unresectable early-stage HCC, and are also recommended for patients at the very early stage (single lesion ≤ 2 cm) [2,5]. In addition, they are applied as palliative treatment for selected patients at later stages in the clinic. Common ablative therapies include chemical ablation, radiofrequency ablation (RFA), microwave ablation (MWA), cryoablation, laser ablation, high-intensity focused ultrasound ablation, and irreversible electroporation, with RFA being most used in the clinic [21,22]. By generating high-frequency alternating current, RFA produces frictional heat of 60–100 °C to initiate the multiple-stage tumor cell destruction, including impairment of tumor cell membrane integrity, mitochondrial dysfunction, inhibited DNA replication, enzyme dysfunction, protein denaturation, dysfunction of RNA synthesis, apoptosis, and vascular injury [21,22,23].

The median OS of curative RFA is approximately 60 months, with the 5- and 10-year OSs being 40–68% and 27–32%, respectively [5,24]. However, the recurrence is relatively high, with a 5-year recurrence rate of 50–81% and a median time to recurrence of 20–30 months [24,25,26]. Complete tumor eradication is independently associated with reduced local tumor progression and improved OS after RFA, and tumor size is the only significant factor that negatively influences the complete response rate [5,26,27,28]. The complete response rate of RFA treatment ranges from 45% to 90%, and a high rate is detected in small lesions (≤2 cm) while a low rate is seen in large lesions [22,29]. Considering the presence of satellite nodules around the primary tumor, an ablation margin of 5–10 mm from the tumor boundaries is suggested [21]. A prospective randomized trial, enrolling 96 patients with small HCC, reported a lower risk of local tumor progression (14.9% vs. 30.2%), a lower risk of intrahepatic recurrence (15.0% vs. 32.7%), and a longer recurrence-free survival (31.7 ± 12.1 vs. 24.0 ± 11.7 months) in the wide margin (≥10 mm) group than in the narrow margin group [30].

To date, evidence that supports the superiority of other ablative techniques to RFA is limited [5,19]. Meta-analyses that compared the efficacy of RFA and percutaneous ethanol injection, a type of chemical ablation, showed the superiority of RFA in terms of OS, complete necrosis rate, local recurrence, and disease-free survival, particularly in patients with a tumor > 2 cm [21,31,32,33]. MWA that uses electromagnetic energy to quickly increase the local temperature to >150 °C has technical advantages over RFA theoretically [22]. However, some large retrospective clinical studies, and randomized studies, failed to show significant differences in OS, local tumor progression, and local recurrence between them [5,34,35,36]. In contrast to RFA and MWA, cryoablation applies repetitive freeze-thaw cycles to create ice crystals in tumor cells and induce tumor cell death, reaching a temperature of <−140 °C [37]. Similarly, the current clinical studies cannot support the superiority of cryoablation to RFA [38]. A multicenter randomized controlled trial enrolled 360 patients with tumors of <4 cm to compare cryoablation and RFA. It reported a lower local tumor progression rate for cryoablation versus RFA (7.7% vs. 18.2%) but similar 1-, 3-, and 5-year OS rates (cryoablation vs. RFA, 97%, 67%, 40% vs. 97%, 66%, 38%) and tumor-free survival (89%, 54%, and 35% vs. 84%, 50%, and 34%) [39]. A propensity-matched population study including 3239 patients with HCC also reported no significant difference in OS between cryoablation and RFA [40]. The efficacy of other ablative techniques is under investigation.

Ablative therapies are assumed to have a synergistic effect on ICI-based immunotherapies [19,41]. Some preclinical studies using mouse tumor models showed that ablative therapies in combination with ICIs significantly increased intratumor infiltration of CD11c+ dendritic cells (DCs), CD4+ T cells, and CD8+ T cells, and decreased the expression of IL-10, resulting in suppressed tumor growth and extended survival of the animals [42,43,44,45,46]. The investigated combination strategies include RFA + anti-PD-1 antibodies, MWA + anti-TIGIT antibodies, MWA + anti-LAG-3 antibodies, and RFA + anti-CTLA4 antibodies. A proof-of-concept clinical trial, enrolling 50 patients with advanced HCC, reported that additional ablation to anti-PD-L1 therapy increased the response rate from 10% to 24%, with the median time to progression, progression-free survival (PFS), and OS being 6.1, 5, and 16.9 months, respectively [47]. A prospective study on 32 patients with advanced-stage HCC confirmed an increase in the intratumoral CD8+ T cell population after RFA combined with tremelimumab, an anti-CTLA4 antibody [48]. However, emerging evidence reveals that insufficient RFA may accelerate the aggregation of immunosuppressive cells and cytokines in the residual tumors, resulting in anti-PD-L1 resistance and tumor progression [49]. Therefore, the safety and efficacy of the combination strategies of ablative therapies and ICIs remain unclear, necessitating further studies for high-level evidence. The ongoing randomized trials are summarized in Table 1.

### 2.2. Transarterial Therapies

Transarterial therapies, including transarterial embolization (TAE), transarterial chemoembolization (TACE), transarterial radioembolization (TARE), and hepatic arterial infusion chemotherapy, are the mainstay treatments for intermediate-stage HCC [2,5,50]. Among them, TACE remains the most applied technique and is considered the standard treatment [5]. Given the distinct blood supply of HCC, where the tumor is fed by the hepatic arteries and approximately 80% of normal liver parenchyma by the portal venous system, TACE can mediate tumor destruction by inducing strong ischemic and cytotoxic effects in tumors without causing serious damage to normal liver parenchyma [51]. Considering the differences in techniques, TACE is subclassified into conventional TACE (cTACE) and drug-eluting bead TACE (DEB-TACE), which administer an emulsion of lipiodol with antitumor drugs and drug-eluting beads, respectively [5,52]. These two techniques can be applied interchangeably, as current evidence suggests similar clinical outcomes between them [19,52,53].

The clinical outcomes of TACE remain unsatisfactory, with a 5-year survival rate of 32.4% [5,54]. The median OS of TACE is approximately 30 months, ranging from 19.4 months in the uncontrolled studies to 37 months in the randomized controlled trials, and the overall ORR is only 52.5% [7,54]. Several combination strategies are being investigated to enhance the initial tumor response, to further improve the clinical outcomes of TACE. Since TACE can synergize the antitumor efficacy of ablative therapies by blocking tumor-feeding arteries, attenuating the “heat-sink effect”, and inducing extended tumor necrosis, this combination treatment is assumed to be promising [5,55,56,57,58,59,60,61].

A randomized trial on 110 patients with intermediate-stage HCC reported the superiority of TACE combined with RFA to TACE alone, in terms of median OS (29 vs. 18 months), median time to progression (TTP; 15.7 vs. 12.4 months), PFS, and best objective response (69.1% vs. 40%) [62]. Some propensity score-matching and retrospective cohort studies revealed similar findings [56,63,64,65]. Although embolizing tumor-feeding arteries can foster tumor necrosis, treatment-induced hypoxia may mediate angiogenesis and significantly attenuate the efficacy of TACE [51]. TKIs with anti-angiogenic effects are therefore proposed to be a synergetic alternative for TACE; however, several phase III randomized controlled trials investigating TKIs as an adjuvant therapy to TACE failed to improve the OS, TTP, or PFS [19,53,66,67,68,69,70,71,72,73].

Despite the advantages of ICI-based immunotherapies [19,20], recent retrospective studies on the combination of TACE and ICIs showed controversial results (Table 1). A study including 142 patients with unresectable HCC demonstrated significantly better clinical outcomes in TACE + pembrolizumab and lenvatinib than in TACE + lenvatinib (median OS, 18.1 months vs. 14.1 months; median PFS, 9.2 months vs. 5.5 months) [74]. Similarly, another retrospective study reported higher disease control rates and longer PFS and OS after TACE + sorafenib and ICIs than after TACE + sorafenib, for intermediate- and advanced-stage HCC, which were 81.82% vs. 55.17%, 16.2 months vs. 7.3 months, and 23.3 months vs. 13.8 months, respectively [75]. Nevertheless, other retrospective studies failed to demonstrate a longer survival in the combination treatment of TACE with ICIs and TKIs. In a retrospective multicenter study involving 323 patients with advanced HCC, the median OS was longer in patients receiving TACE + nivolumab than in those receiving nivolumab monotherapy (35.1 months vs. 16.1 months), but the difference was not significant [76]. In addition, a multicenter, retrospective, cohort study on 534 patients with intermediate- and advanced-stage HCC showed that TACE + camrelizumab and apatinib was superior to TACE alone giving an improved median PFS (13.7 months vs. 7.0 months), ORR (55.9% vs. 36.8%), and grade 3–4 adverse event rate (17.6% vs. 2.9%); however, no statistical difference in OS was found [77]. Further randomized trials are needed to clarify the efficacy of TACE combined with ICIs.

## 3. Cytokines in the Combined Strategies of Locoregional Interventional Therapies and ICI-Based Immunotherapy: Role and Therapeutic Potentials

### 3.1. Different Roles of Complete and Incomplete LITs in Modulating Antitumor Immune Response and Tumor Progression

LITs combined with ICI-based immunotherapies represents a rapidly evolving field in the management of HCC, and better understanding the biological mechanisms for LITs and ICI-based immunotherapies is imperative for refining treatment protocols. LITs are assumed to synergize with ICIs to improve the clinical outcomes of patients with HCC by enhancing antitumor immune responses [19,20]. The immunostimulatory effects of LITs have been gradually elucidated, which include [19]: (1) promoting the release of tumor antigens; (2) modulating the expression of ICPs, including PD-1, PD-L1, CTLA-4, LAG-3, etc.; (3) producing immunomodulatory cytokines and chemokines, including INF-γ, IL-2, CXCL-9, etc.; and (4) facilitating the intratumoral infiltration of tumor-killing immune cells, including cytotoxic T cells, NK cells, and Th1 T cells. However, accumulating evidence suggests that complete LITs can promote antitumor immune response, whereas incomplete LITs may inhibit antitumor immune response and hinder the efficacy of ICIs, given their immunosuppressive and tumor-promoting effects.

Incomplete LITs may lead to insufficient tumor eradication and are not rare in clinical practices. For thermal therapies, irreversible cellular injury occurs within several minutes at 50–54 °C or <−20 °C, and rapid coagulative necrosis occurs at ≥60 °C [19,21,23]. Complete tumor eradication occurs only when the temperature of the entire tumor, with a satisfactory ablation margin, reaches the cytotoxic level [21,22]. However, the Joule Effect, makes heat deposition vary a lot within the ablated tumor, with a higher temperature in the central zone and a lower one in the peripheral and surrounding zones [23]. Tumor cells in the central zone may suffer a cytotoxic temperature and be eradicated [23]. However, those in the peripheral zone undergo sublethal heating, which may enable tumor cells to survive and acquire aggressive characteristics, leading to tumor recurrence and progression [23,49,78,79]. Meanwhile, owing to the intrinsic limitations of the techniques, transarterial therapies are considered palliative that barely leads to complete tumor destruction [5,50]. To date, tumor size is considered the key independent factor for complete tumor eradication for LITs. The larger the size, the less likely the complete tumor cell destruction is, and vice versa [2,5,19]. In addition, the efficacy of thermal ablation is affected by the location of the tumor, for example, lesions in the subdiaphragmatic and subcapsular regions and those close to the intrahepatic and vascular structures are less likely to be completely eradicated [22].

In contrast to the immunostimulatory and tumor-killing roles of complete LITs, incomplete LITs may contribute to an immunosuppressive tumor microenvironment and tumor progression. Several animal studies and histological analyses of patients with HCC demonstrated significant infiltration of MDSCs, TAMs, and Treg cells in both the tumor and peripheral blood after incomplete LITs, but reduced cytotoxic T cells, NK cells, Th1 cells, and DCs [49,78,80,81]. Additionally, incomplete LITs were found to increase the population of the activated myofibroblasts in the residual tumor, and these myofibroblasts subsequently function to synthesize and release immunoinhibitory cytokines [82] along with other tumor-promoting immune cells. Remarkably, a preclinical study, using an orthotopic HCC murine model, showed that incomplete thermal ablation could elicit tumor resistance to PD-1 blockade by inhibiting T cell function and proliferation by mediating Type 2 macrophage programming [49]. Moreover, incomplete LITs result in a more significant and prolonged hypoxic and ischemic microenvironment than do complete LITs, contributing to higher expressions of hypoxia-related molecules, including HIFs and HSPs [83,84,85,86,87]. Hypoxia-related molecules are the major mediator for modulating tumor angiogenesis, tumor cell growth, invasion, and migration [87,88]. Taken together, incomplete LITs are less likely to synergize with ICI-based immunotherapies, theoretically.

### 3.2. Roles of Cytokines in Modulating Local and Systemic Responses to LITs

Cytokines, the major molecular regulators of the innate and adaptive immune response [89,90], play crucial roles in modulating intratumoral and systemic responses to LITs. Overall, LITs can destruct tumor cells to trigger inflammatory responses and tumor antigen-mediated immune responses (Figure 1) [19,20]. Inflammatory responses occur at the early stage after LITs, increasing the permeability of tumor vessels and promoting the injured tumor cells to produce chemokines (e.g., CCL2, CCL3, CCL4, CCL8, etc.) and proinflammatory cytokines (e.g., TNF-α, IL-1, IL-6, IL-17, etc.). Then, the inflammatory cells (e.g., neutrophils, macrophages, and Th17 cells, etc.) are attracted to infiltrate the ablated tissues and subsequently secrete more proinflammatory cytokines, immunomodulatory cytokines (e.g., IFN-γ, IL-2, IL-12, etc.), and chemokines [91,92,93,94,95,96]. Along with the immunomodulatory cytokines secreted by proinflammatory cells, LIT-induced tumor antigens can trigger the activation, proliferation, and maturation of antitumor immune cells, DCs, Th1 cells, cytotoxic T cells, and NK cells [42,91,97,98,99,100]. Under the guidance of chemokines, antitumor immune cells migrate to the tumor to kill tumor cells. In addition, LIT-induced HIFs and HSPs are significant mediators for the subsequent inflammatory and immune responses [84,101,102,103,104,105]. HIFs mediate the inflammatory responses by promoting the activation and intratumoral infiltration of inflammatory cells and the production of proinflammatory cytokines [100,106,107]. Meanwhile, as a multifunctional modulator, HSPs are also involved in modulating inflammatory reactions, and HSP70 is considered the most promising inducer of inflammation by activating monocyte and promoting the production of proinflammatory cytokines [88,104].

Mechanically, inflammation occurs immediately after the procedure and gradually attenuates, followed by antitumor immune responses that are dominant at a later stage [105,108,109]. However, a prolonged inflammatory response has been observed in patients after incomplete LITs, contributing to the formation of an immune suppressive microenvironment (Figure 1) [49,78,94,105,109]. So far, the mechanisms for the prolonged inflammation induced by incomplete LITs have been only partially unveiled. As previously mentioned, incomplete LITs lead to a higher expression of both HIFs and HSPs, which subsequently prolong the inflammatory responses systemically and locally [101,105,110]. In addition, COX-2, a membrane-bound molecule expressed by both tumor cells and immune cells, is also upregulated by incomplete LIT-induced HIFs and proinflammatory cytokines, to enhance the inflammatory responses in the residual tumor [83]. The prolonged inflammation can not only promote the production of immunosuppressive cytokines (e.g., IL-6, IL-10, TGF-β, VEGF, HGF, CCL2, etc.) but also enhance the intratumoral infiltration of immunosuppressive cells (e.g., TAMs, MSDCs, and Tregs Cells). As a result, the immunosuppressive microenvironment is formed, antitumor immunity is inhibited, and tumor survival and progression occur in the residual tumor (Figure 1) [49,78,80,111].

#### 3.2.1. Interferons (IFNs)

IFNs are a group of glycoproteins mainly produced by a variety of immune cells in response to the presence of antigens or pathogens [112,113]. It is well-known that IFNs play crucial roles in modulating innate and adaptive immunity through logical actions that mainly include: (1) activating T cells and B cells; (2) promoting T cell proliferation; (3) inducing the expression of MHC class molecules on tumor cells; (4) promoting the maturation of DCs; and (5) indicating macrophages [112,113,114]. The IFN family mainly includes IFN-α, IFN-β, and IFN-γ, among them, IFN-γ, an immunostimulatory cytokine, has been reported to engage in the immune responses in cancer patients after LITs (Table 2) [114]. The elevated expression of IFN-γ in both peripheral blood and tumor was observed in some animal studies and clinical studies [42,91,97,115,116]. In a murine HCC model, mice treated with complete RFA displayed upregulation of IFN-γ in the peri-ablation liver tissues, subsequently contributing to the increase in local infiltrations of both CD169+ macrophage and CD8+ T cells [91]. Meanwhile, other animal studies also reported significantly increased intratumoral expression of IFN-γ after complete ablation that enhanced the cytotoxicity of CD8+ T cells [42,97]. In addition, LIT-induced IFN-γ could also upregulate the expression of PD-L1, indicating the potential of LITs to synergize with ICIs [42].

#### 3.2.2. Interleukins (ILs)

ILs are a group of cytokines that engage in a variety of physiological and pathological processes, which include but are not limited to, inflammation, innate immunity, adaptive immunity, and tumor initiation and progression [146]. The roles of ILs in the responses to LITs have been gradually elucidated (Table 2). Remarkably, immunostimulatory ILs are upregulated in patients who received LITs; however, higher expression levels of immunoinhibitory interleukins were noted after incomplete LITs but not after complete LITs [78]. Some clinical studies investigating the systemic responses to LITs reported that LITs significantly increased the serum levels of IL-2 and IL-12, which both are major mediators for the activation, proliferation, and infiltration of tumor-killing immune cells, indicating the activation of antitumor immunity after LITs [116,147,148,149]. Meanwhile, complete RFA could predominantly upregulate the expression of IL-7, a major factor for T cell differentiation and function, in the periablational zone, to enhance the subsequent T cell infiltration and activation [91]. Besides, the expression of IL-17, a known proinflammatory cytokine, was elevated in both the peripheral blood and tumor after LITs, suggesting their engagement in responses to LITs [115,116]. Further studies are required to detail their roles in LITs.

To date, there is emerging evidence also illustrating the roles of some ILs, including IL-6 and IL-10, in inhibiting antitumor immune responses and promoting tumor progression after incomplete ILTs [78]. IL-6, a pleiotropic cytokine, is well-known for its pro-inflammatory and tumor-promoting effects [150]. LITs were found to significantly increase the expression of IL-6 in both peripheral blood and targeted tumor within hours postoperatively, however, incomplete LITs showed a higher and prolonged expression of IL-6 than did complete LITs [80,93,105,118,119,120,121,122,123,145,151,152,153,154,155,156]. After incomplete LITs, IL-6 was produced by injured tumor cells and infiltrated inflammatory cells in the residual tumor, and IL-6 can in turn foster the activation and infiltration of inflammatory cells as well as the production of proinflammatory cytokines, resulting in a prolonged inflammation [80,122,152]. In addition, IL-6 could also promote tumor cell survival, growth, proliferation, transformation, invasion, and migration by upregulating TGF-β and activating the STAT3 and HGF/c-MET pathways [80,120,121,122,156,157]. Furthermore, IL-6 was also found to induce the production of VEGF and facilitate angiogenesis in the residual tumor, leading to tumor progression [121]. Remarkably, evidence from some prospective trials demonstrated the association between postoperative increases of IL-6 in serum and poor tumor response, as well as shorter PFS in LIT-treated HCC [118,151,152]. As a well-known immunosuppressive cytokine, IL-10 may also engage in promoting tumor progression after incomplete LITs. Its elevated serum level was found in patients suffering poor survival after TACE [125]. Besides, in vitro and in vivo preclinical studies demonstrated that incomplete RFA significantly upregulated IL-10 expression, leading to an inhibited immune response in the residual tumor [78].

#### 3.2.3. Chemokines

As the essential mediators for immune cell infiltrations and tumor cell migration, chemokines are found to play crucial roles in modulating the responses to LITs [158]. Similarly, complete and incomplete LITs have different impacts on stimulating the production of chemokines. Complete ablation predominantly elevates the expression of CCL3, CCL4, and CXCL14 in the periablational liver tissue, enhancing the infiltration of activated NK cells, CD4+ T cells, and CD8+ T cells [91,127,128]. In contrast, incomplete LITs can upregulate CCL8 and CCL2 to enhance the intratumoral infiltration of immunosuppressive cells (i.e., TAMs, Tregs, and MDSCs), contributing to the inhibition of antitumor immunity [78,159]. In addition, a prospective study discovered elevated serum levels of macrophage MIF in patients with intermediate-stage HCC 1 day after TACE, and its association with poor prognosis needs to be further detailed [160].

#### 3.2.4. TNF-α

TNF-α is a crucial proinflammatory cytokine that promotes tumor progression by modulating tumor-associated inflammatory responses [161]. Evidence from the existing studies has suggested that LITs could upregulate TNF-α to drive inflammatory responses, immune responses, and tumor progression [42,78,91,97,110,118,129,156]. Current preclinical studies and clinical studies have reported an elevated expression of TNF-α in both peripheral blood and the treated tumor within hours after LITs, suggesting systemic and local responses to LITs. Meanwhile, TNF-α was found to participate in the incomplete RFA-induced infiltration and activation of TAMs [78]. In a murine HCC study, incomplete RFA initially stimulated residual tumor cells to produce CCL2 through the upregulation of TNF-α, contributing to the infiltration and activation of TAMs [78]. The activated TAMs in the residual tumor in turn produced more TNF-α and CCL2, resulting in enhanced inflammation [78].

#### 3.2.5. TGF-β

TGF-β, a well-known immunosuppressive and tumor-promoting cytokine, is upregulated by residual tumor cells after incomplete LITs and engages in the subsequent immune inhibition and tumor progression [80,111,162,163,164]. In a murine HCC model, sublethal heating predominantly promoted TGF-β production by upregulating METTL1 in tumor cells, resulting in the increased accumulation of MDSCs, reduced CD8+ T cell population, and enhanced tumor growth and metastasis [81]. In addition, insufficient heating also upregulates NEDD4 to enhance TGF-β production and TGF-β–mediated tumor progression [131]. Hypoxia can also induce overexpression of TGF-β after incomplete LITs [84]. A murine HCC study reported that sublethal heating enabled tumor cells to acquire enhanced proliferative, invasive, and metastatic characteristics via the hypoxia/HIF-1α TGF-β/EMT axis [84]. Remarkably, a preclinical study on a murine HCC model demonstrated that suppressing the TGF-β pathway significantly enhanced the anticancer effects of PD-1 blockade combined with RFA [43].

#### 3.2.6. VEGF

VEGF is the key promoter for angiogenesis [165] that engages in the local and systemic response to LITs. Many preclinical and clinical studies have demonstrated upregulated VEGF in patients with HCC after receiving LITs [86,87,101,105,132,133,134,135,136,137,138,139,140,156,164,166,167,168,169,170]. Compared to complete LITs, incomplete LITs produce a higher expression of VEGF. HIF-1α has been acknowledged to participate in the incomplete LIT-induced overexpression of VEGF and the VEGF-induced angiogenesis via the PI3K/Akt/HIF/VEGF pathway and HIF/VEGF/EphA2 pathway, respectively [86,87,135,171]. In addition to its pro-angiogenic role, VEGF can also elicit the stemness of tumor cells and promote the survival and treatment resistance of tumor cells [141]. Meanwhile, an elevated serum level of VEGF was observed in patients with HCC after LITs, especially when large tumors were present [135,138,167]. Thus, a high level of VEGF was found to be associated with, as well as an independent prognostic factor for, a poorer prognosis [142,167,169,172].

#### 3.2.7. HGF

HGF plays a crucial role in promoting tumor progression, including the complete LIT-driven one, through the activation of the -MET pathway [157,173]. Both hypoxia and sublethal hyperthermia after ablation were found to stimulate tumor cells and stromal cells to produce HGF in the residual tumor [93,105,123,155,156]. Interestingly, MWA using low power led to a higher level of HGF compared to using high power [105]. In addition, a high level of HIF-1α after incomplete LITs also contributed to the elevated expression of HGF in the residual tumor, but HGF could in turn synergize with HIF-1α to promote tumor progression [82,174]. Meanwhile, several preclinical studies also revealed that incomplete ablation could promote tumor progression and tumor angiogenesis via the IL-6/HGF/c-Met pathway, the HGF/c-Met/STAT3 pathway, and the upregulation of VEGF [82,121].

### 3.3. Specific Cytokines as Therapeutic Targets for Reducing ICI-Induced Immune-Related Adverse Event (irAEs)

ICI-based immunotherapies lead to tumor destruction by renormalizing antitumor immune responses. However, ICPs’ role in maintaining immune homeostasis may cause irAEs [10]. The incidence of irAEs varies by the treatment protocols and types of tumors [10,175]. For HCC, the incidences of severe irAEs for PD-1/PD-L1 blockade, CTLA-4 blockade, dual blockade of PD-L1/PD-L1 and CTLA-4, and ICIs combined with TKIs are 10–20%, ~25%, ~50%, and ~67%, respectively. ICI-induced irAEs mainly include skin toxicities, diarrhea, colitis, hepatitis, and pneumonitis. Among them, diarrhea is the most common ICI-induced irAEs for HCC, and the incidence ranges from 10% to 43%. Remarkably, the incidence of hepatitis is higher in HCC than in other types of tumor (9–14% versus 1–9%). Cytokines are also contributing factors for ICI-induced irAEs. Several prospective trials and retrospective studies showed that the serum level of IL-6 is higher in patients with colitis after ICI treatment, and that it was a significant and independent risk factor for irAEs [176,177,178]. Upregulation of TNF-α was found to be associated with ICI-induced gastritis or colitis in some retrospective studies. Furthermore, some cancer patients with CTLA-4 treatment-related colitis showed high serum concentrations of IL-17, and a preclinical animal study further confirmed this finding and found that CTLA-4 blockade promoted Th17 T cell differentiation.

Although glucocorticoids are the current standard of care for ICI-induced irAEs, they are not always effective, but may attenuate the antitumor efficacy of ICIs. Interestingly, current clinical studies have unveiled the advantages of agents targeting specific cytokines in managing irAEs after failed glucocorticoids therapy. In a retrospective study including 34 patients who suffered nivolumab-induced grade 3/4 irAEs, tocilizumab, an anti-IL-6 receptor monoclonal antibody, provided clinical improvements in 27 patients (79.4%) [179]. Other prospective trials showed similar benefits with IL-6 blockade for refractory irAEs induced by ICI treatment [178,180]. Another retrospective analysis of 29 patients with metastatic melanoma identified that most patients that had experienced failed corticosteroids responded to infliximab (21/29) [181]. Some case studies showed that the blockade of IL-17 may serve as an alternative in the management of refractory ICI-related irAEs [182]. Taken together, specific cytokines could be promising therapeutic targets for reducing the irAEs induced by ICI-based immunotherapy.

## 4. Future Directions

With the development of ICI-based immunotherapies, interest in the combination of LITs and ICI-based immunotherapies for the management of HCC is also growing. The efficacy of LITs combined with ICIs for managing HCC is controversial according to existing retrospective data. However, ongoing prospective trials that aim to investigate the combination therapies will further answer whether these combinations better benefit patients suffering from HCC in the coming years. Emerging evidence has reshaped our understanding of the roles of LITs in cancer, complete LITs may enhance antitumor responses, but incomplete LITs may result in immune suppression and tumor progression. Some cytokines have been identified as playing roles in modulating the local and systemic responses to incomplete LITs, but most of the mechanisms for LITs remain unclear. Since a better understanding of the biological processes behind LITs and ICIs-based immunotherapies can help to refine the treatment protocols, more mechanistic research is required. In addition, despite the current findings of the promising potentials of targeting specific cytokines in managing refractory irAEs, most of them were from retrospective or case studies; therefore, prospective trials are urgently needed for verification.

## 5. Conclusions

LITs are believed to enhance antitumor immunity by promoting the release of tumor antigens and proinflammatory cytokines; however, LITs may result in immune suppression and tumor progression by upregulating proinflammatory and immunosuppressive cytokines and promoting the accumulation of suppressive cells. In addition, high levels of cytokines are found to be associated with ICI-related irAEs, and targeting specific cytokines could benefit refractory irAEs. Taken together, the rediscovery of the roles of incomplete LITs in tumors suggests that the addition of agents targeting specific cytokines to the combination therapies of LITs and ICIs should be considered to improve the therapeutic efficacy and reduce treatment-related irAEs.

## Figures and Tables

**Figure 1 cancers-15-01324-f001:**
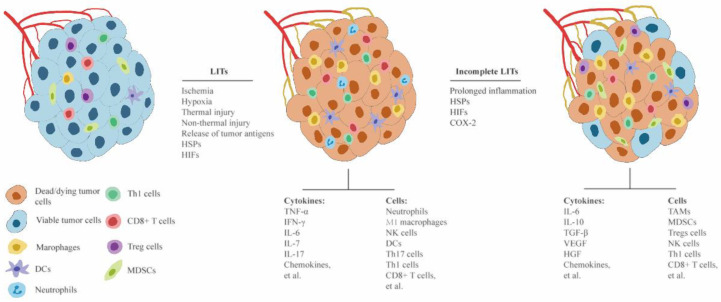
Inflammatory and immune responses to LITs. LITs can destruct tumor cells by multiple mechanisms, including thermal damage, non-thermal damage, and induction of a hypoxic and ischemic microenvironment. In response to LITs, inflammatory cells are initially stimulated to infiltrate into the ablated tumor tissue to eliminate the dying or dead tumor cells. Subsequently, inflammatory cells also produce more proinflammatory cytokines and immunomodulatory cytokines to mediate the following antitumor immune responses. Remarkably, incomplete LITs can induce significant and prolonged inflammation in the residual tumor tissue. Together with a high level of HIFs, HSPs, and COX-2, the prolonged inflammation remarkably can lead to the upregulation of immunosuppressive cytokines and tumor-promoting cytokines, and enhanced accumulation of immunosuppressive cells, resulting in the formation of a tumor-stimulating microenvironment and the inhibition of antitumor immune response.

**Table 1 cancers-15-01324-t001:** Selected ongoing phase III clinical trials investigating local interventional therapies combined with immune checkpoint inhibitor-based immunotherapies for HCC.

Sponsor	Acronym	Intervention	Population	Sample Size	Primary Endpoints	Expected End	Trial Registration ID
Merck Sharp & Dohme LLC (Rahway, NJ, USA)	LEAP-012	TACE plus lenvatinib plus pembrolizumab versus TACE plus placebo	Intermediate stage HCC	450	OS and PFS	31 December 2029	NCT04246177
Hoffmann-La Roche (Basel, Switzerland)	-	Atezolizumab plus bevacizumab plus TACE versus TACE	Intermediate stage HCC	342	PFS and OS	28 February 2029	NCT04712643
AstraZeneca (Cambridge, UK)	EMERALD-1	Durvalumab plus bevacizumab plus TACE versus Durvalumab plus TACE versus TACE plus placebo	Locoregional HCC	724	PFS	19 August 2024	NCT03778957
AstraZeneca	EMERALD-3	Tremelimumab plus durvalumab plus lenvatinib plus TACE versus tremelimumab plus durvalumab plus TACE versus TACE	Locoregional HCC	525	PFS	29 January 2027	NCT05301842
Bristol-Myers Squibb (New York, NY, USA)	CheckMate 74W	Nivolumab plus ipilimumab plus TACE versus nivolumab plus TACE versus TACE	Intermediate HCC	26	TTTP and OS	29 January 2024	NCT04340193
The Clatterbridge Cancer Centre NHS Foundation Trust (Birkenhead, UK)	TACE-3	Nivolumab plus TACE/TAE versus TACE/TAE	Intermediate stage HCC	522	OS and TTTP	June 2026	NCT04268888
Jiangsu HengRui Medicine Co., Ltd. (Lianyungang, China)	-	Camrelizumab plus Apatinib plus versus TACE	Incurable HCC	360	PFS	30 July 2026	NCT05320692
Zhongda Hospital	-	Penpulimab plus anlotinib plus TACE versus penpulimab plus anlotinib	Advanced stage HCC	109	PFS	31 March 2024	NCT05344924
AstraZeneca	EMERALD-2	Curative therapy (resection of ablation) plus durvalumab plus bevacizumab versus Curative therapy (resection of ablation) plus durvalumab versus Curative therapy (resection of ablation) plus placebo	Early/intermediate stage HCC	908	RFS	31 May 2024	NCT03847428
Merck Sharp & Dohme LLC	KEYNOTE-937	Curative therapy (resection of ablation) plus Pembrolizumab versus Curative therapy (resection of ablation) plus placebo	Early/intermediate stage HCC	950	RFS	31 August 2029	NCT03867084
Bristol-Myers Squibb	CheckMate 9DX	Curative therapy (resection of ablation) plus nivolumab versus Curative therapy (resection of ablation) plus placebo	Early/intermediate stage HCC	545	RFS	16 December 2025	NCT03383458
Hoffmann-La Roche	IMbrave050	Curative therapy (resection of ablation) plus atezolizumab plus bevacizumab versus none	Early/intermediate /advanced stage HCC	668	RFS	16 July 2027	NCT04102098

TTTP: time to TACE progression; RFS: recurrence-free survival.

**Table 2 cancers-15-01324-t002:** Cytokines involved in the local and systemic responses to locoregional interventional therapies.

Cytokine	Procedure	Location	Pathological Actions	Ref.
IFN-γ	TACE, ablative therapies	Intratumor, peripheral blood	Cytotoxic T cells activation and function; Macrophage activation; Upregulation of PD-L1 expression	[42,91,97,117]
IL-6	TAE; ablative therapies	Intratumor; peripheral blood	Prolonged inflammation, tumor cell undergoing EMT; tumor cell proliferation, invasion, and migration; angiogenesis	[80,105,118,119,120,121,122,123,124]
IL-7	Ablative therapies	Intratumor	T cell infiltration and activation	[91]
IL-10	Ablative therapies	Intratumor; peripheral blood	Intratumor; peripheral blood	[125]
IL-17	TAE	Intratumor	Inflammation	[126]
CCL2	Ablative therapies	Intratumor	Monocyte and TAM infiltration; prolonged inflammation	[78]
CCL8	Ablative therapies	Intratumor	TAM infiltration	[91]
CXCL14	Ablative therapies	Intratumor	Intratumor	
CCL3	Ablative therapies	Intratumor	CD4+ T cell and CD8+ T cell infiltration	[91,127,128]
CCL4	Ablative therapies	Intratumor	CD4+ T cell and CD8+ T cell infiltration	[91,127]
TNF-α	Ablative therapies	Intratumor; peripheral blood	Inflammation; CCL2 induction	[97,117,124,129,130]
TGF-β	TAE; ablative therapies	Intratumor	MSDCs infiltration; inhibition of CD8+T cell infiltration; tumor cell undergoing EMT; tumor cells survival, proliferation, invasion, and migration	[43,80,81,84,111,131]
VEGF	TACE; ablative therapies	Intratumor; peripheral blood	Angiogenesis; tumor cell stemness	[85,86,87,101,105,121,123,132,133,134,135,136,137,138,139,140,141,142,143,144]
HGF	Ablative therapies	Intratumor	Tumor cell proliferation, invasion, and migration	[82,105,121,122,123,145]

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
