# Peer review of "How Biology Guides the Combination of Locoregional Interventional Therapies and Immunotherapy for Hepatocellular Carcinoma: Cytokines and Their Roles"

_cancers, 2023, doi:10.3390/cancers15041324_

Round 1

Reviewer 1 Report

This is a detailed literature review on the role of cytokines on patients with HCC undergoing loco-regional therapies and receiving adjuvant ICIs. The topic is of high interest. The manuscript is well organized although some sections are too long, hampering the reading. The authors are kindly invited to consider the following comments:  

-         I would recommend to shorten section 2. I understand that the authors aim to provide an overview of loco-regional therapies for those readers not familiar with HCC management. This may be necessary but I felt that this section could still met this purpose after significant shortening.

-          In section 3.1 it can be read: “Current clinical studies failed to demonstrate the consistent superiority of LITs combined with ICI-based immunotherapies in the management of HCC…).” As shown in table 1, there is a great number of ongoing phase III trials and the available finished studies is reduced to support this statement. I would rather say that this is a rapidly evolving field in which current uncertainties will soon be elucidated, or I would soften significantly the statement including also one or two negative studies to support it.

-          In opinion of the authors, Could dynamic changes of cytokines be used as biomarkers to predict tumor response to loco-regional therapies in order to select candidates to receive adjuvant therapies? Is there any evidence on this?

-          In table 1, Could you provide some more data from the RCT such as the sponsor, sample size and expected end of the study?

-          Figure 1 is not readable. Please improve quality or increase the font size.

-          I would re-schedule sections 4 and 5 so that the future directions are presented before the conclusions of the review. The last section of the manuscript should be the conclusions.  

Author Response

Point 1: I would recommend to shorten section 2. I understand that the authors aim to provide an overview of loco-regional therapies for those readers not familiar with HCC management. This may be necessary but I felt that this section could still met this purpose after significant shortening.

Response 1: Thank you for your comment. Section 2 has been shortened as suggested.

Point 2: In section 3.1 it can be read: “Current clinical studies failed to demonstrate the consistent superiority of LITs combined with ICI-based immunotherapies in the management of HCC…).” As shown in table 1, there is a great number of ongoing phase III trials and the available finished studies is reduced to support this statement. I would rather say that this is a rapidly evolving field in which current uncertainties will soon be elucidated, or I would soften significantly the statement including also one or two negative studies to support it.

Response 2: Thank you for your suggestion. This description has been revised accordingly.

Point 3: In opinion of the authors, Could dynamic changes of cytokines be used as biomarkers to predict tumor response to loco-regional therapies in order to select candidates to receive adjuvant therapies? Is there any evidence on this?

Response 3: Thank you for your question. Some prospective trials demonstrated that the postoperative upregulation of both IL-6, IL10, and VEGF in serum were associated with poor prognosis and could thus be considered independent biomarkers to predict tumor response to loco-regional therapies for HCC (PMID: 28787261, 24035756, 31249138, 31751357, 35153515, 19016764, 26019468, 27822426, 18177453). These findings indicated that the dynamic change of these cytokines may theoretically be applied to select candidates to receive adjuvant therapies. However, further clinical studies are required to clarify it. Relevant information about the prognostic value of IL-6 in loco-regional therapies for HCC has been added to section 3.2.2.

Point 4: In table 1, Could you provide some more data from the RCT such as the sponsor, sample size and expected end of the study?

Response 4: Thank you for your comment. The sponsors, sample size, and expected ends of the RCTs have been added to Table 1 as suggested.

Point 5: Figure 1 is not readable. Please improve quality or increase the font size.

Response 5: Thank you for pointing it out. The quality of Figure 1 has been improved and inserted in the manuscript.

Point 6: I would re-schedule sections 4 and 5 so that the future directions are presented before the conclusions of the review. The last section of the manuscript should be the conclusions.

Response 6: Thank you for your suggestion. These two sections have been re-scheduled as recommended.

Reviewer 2 Report

This is an exhaustive and well conducted review concerning an interesting topic. I have no commentaries.

Author Response

Thank you for you comments.

Reviewer 3 Report

Dear Dr. 

Editor, 

Overall recommendation: 

 Accept 

Final comments:

   The authors have shown recent strategies of advanced hepatocellular carcinoma including immune checkpoint inhibitor (ICI) – based and locoregional interventional therapies. They have shown very attractive knowledge about how ICI demonastrate anti-HCC effects

  I think their data is confident and good for publication in the present form.

Kansai Medical University

Katsunori Yoshida

Author Response

Thank you for your comment.

Round 2

Reviewer 1 Report

The authors have successfully addressed all my initial concerns. The manuscript is much improved in its present version.